# The influence of land use and management on the behaviour and persistence of soil organic carbon in a subtropical Ferralsol

Laura Hondroudakis[1], Peter M. Kopittke[1,*], Ram C. Dalal[1], Meghan Barnard[1], Zhe H. Weng[1,2]

[1]The University of Queensland, School of Agriculture and Food Sustainability, St Lucia, Queensland 4072, Australia
[2]The University of Adeliade, School of Agriculture, Food, and Wine, Urrbrae, South Australia 5064, Australia

*Correspondence to*: Peter M. Kopittke (p.kopittke@uq.edu.au)

**Abstract.** A substantial carbon (C) debt has been accrued due to long-term cropping for global food production emitting carbon dioxide from soil. However, the factors regulating the persistence of soil organic C (SOC) remain unclear, with this hindering our ability to develop effective land management strategies to sequester C in soil. Using a Ferralsol from semi-arid subtropical
Australia, alteration of bulk C contents and fractions due to long-term land use change (up to 72 y) were examined with a focus on understanding whether SOC lost due to cropping could be restored by subsequent conversion back to pasture or plantation. It was found that use of soil from cropping for 72 y resulted in the loss of > 70% of both C and N contents. Although conversion of cropped soil to pasture or plantation for up to 39 y resulted in an increase in both C and N, the C content of all soil fractions were not restored to the original values observed under remnant vegetation. The loss of C with cropping was most pronounced
from the particulate organic matter fraction, whilst in contrast, the portion of the C that bound strongly to the soil mineral particles (i.e., the mineral-associated fraction) was most resilient. Indeed, microbial-derived aliphatic C was enriched in the fine fraction of mineral-associated organic matter (< 53 µm). Our findings were further confirmed using synchrotron-based micro-spectroscopic analyses of intact microaggregates which highlighted that binding of C to soil mineral particles is critical to SOC persistence in disturbed soil. The results of the present study extend our conceptual understanding of C dynamics and
behaviour at the fine scale where C is stabilised and accrues, but it is clear that restoring C in soils in semi-arid landscapes of subtropical regions poses a challenge.

## 1 Introduction

The release of greenhouse gas (GHG) emissions from anthropogenic sources has increased rapidly over the last few decades despite rigorous international efforts to reduce emissions. Globally, soil organic carbon (SOC) stocks have been reduced by
20-60% with land use change to long-term cropping (Kopittke et al., 2017), with land use change having released an estimated 116 Pg of carbon (C) to the atmosphere (Sanderman et al., 2017). However, this current C debt also corresponds to an opportunity to sequester atmospheric C (Sykes et al., 2020). Soils are second only to the oceans in their C storage size, and the amount of C stored in the atmosphere (875 Gt C) and vegetation (450 Gt C) combined is less than that in soils (1700 Gt C outside of permafrost regions) (Friedlingstein et al., 2022).

SOC positively influences soil water, structure, health, and plant productivity, and protecting and building SOC is essential to climate change mitigation (Lehmann et al., 2020a; Oldfield et al., 2019). To optimise SOC sequestration, the complex behaviour of C in soils, and the factors which control its dynamics, need to be understood. Lehmann et al. (2020b) proposed that functional complexity from the interactions between molecular diversity (i.e., C forms), spatial heterogeneity (i.e.,
distribution of C forms) and temporal variability in the soil system regulate the persistence of SOC. The first of these factors, molecular diversity, is important because a higher diversity of molecules will increase the metabolic cost for micro-organisms past a certain threshold as large molecules are more difficult to assimilate. The second of these factors, spatial heterogeneity, refers to the spatial distribution of C forms and the associated likelihood of decomposers to access this SOC due to adsorption of SOC to mineral surfaces and metal ions, and occlusion within aggregates. The final factor, temporal variability also

influences the persistence of SOC as changes in conditions such as nutrients, temperature, and moisture, impact the activities of microbes, resulting in changes to decomposition. How these three factors, molecular diversity, spatial heterogeneity, and temporal variability, interact and the influence that this and land use change have on SOC persistence is only poorly understood (Lehmann et al., 2020b). This is especially true in tropical and subtropical soil, with many previous studies having been conducted in temperate systems such as in Europe and North America. Furthermore, there is uncertainty regarding the composition and formation of aggregates and mineral-associated SOC and subsequently how these can be influenced by land management practices (Angst et al., 2021).

The objective of the present study was to improve our understanding of the impact of long-term land use change on SOC and the underlying factors regulating its persistence by examining both the molecular diversity and spatial heterogeneity of SOC. Specifically, the aims were three-fold: to investigate the impacts of land use change on bulk C contents and pools, the shift of C forms, and the fine scale distribution of C forms. A Ferralsol from subtropical southeast Queensland, Australia, was used and four land uses were compared: (i) undisturbed soil (remnant vegetation) (ii) a cropped soil, (iii) pasture, and (iv) plantation. First, the alteration of bulk C contents due to land use change were quantified. Next, the properties of this SOC were compared between land uses, with this providing critical information on the properties regulating SOC persistence – for this, changes in microbial respiration and fractionation were examined. In addition, synchrotron-based soft X-ray spectroscopy (SXR) was used to identify the C forms within each land use. Finally, to improve our understanding regarding the fine scale distribution of C, the lateral distribution of C forms was compared between land uses using synchrotron-based infrared microspectroscopy (IRM) analyses. The information obtained from these analyses contributes to the emerging body of evidence that soil functional complexity substantially influences C turnover and persistence.

## 2 Materials and methods

### 2.1 Soil collection and study site

Soil samples were collected from Kingaroy in south-east Queensland, Australia (26.7°S, 151.8°E). Using the World Reference Base, this soil is classed as a Ferralsol (being a Ferrosol in the Australian Soil Classification) and has developed on basalt flows that have been intensely weathered. The site is in a lower rainfall environment (676 mm y$^{-1}$) and has a slope of < 5 %. Soil was collected from four land uses which were all within 700 m of one another (Fig.1): (a) remnant vegetation, (b) land which was converted from remnant vegetation to cropping (peanut-maize cultivation) for 33 y, then converted to pasture for 18 y before being converted to plantation (*Corymbia citriodora* subspecies variegata) for the last 21 y ("plantation"), (c) land which was converted from remnant vegetation and has been used for cropping (peanut-maize cultivation) for the last 72 y ("cropped"), and (d) land which was converted from remnant vegetation to cropping (peanut-sorghum-maize cultivation) for 33 y and then grazed pasture for the last 39 y ("pasture") (Supplementary Table S1 and Supplementary Fig. S1; Zhang et al. (2020)). Within each land use, five replicate samples were taken at random points within 10 m of each other, with soil collected to a depth of 0-10 cm. As noted by Zhang et al. (2020) for this experimental system, this depth of sampling (0-10cm) corresponds to the depth of soil disturbance by tillage, with the cropped soils having been tilled by conventional tillage practices since their conversion from native vegetation. Although the cropped soils have had fertiliser applied, no fertilisers have been applied to the pasture or plantation soils (Supplementary Table S1). No lime has been applied. Further information on the soil properties is provided by Zhang et al. (2020).

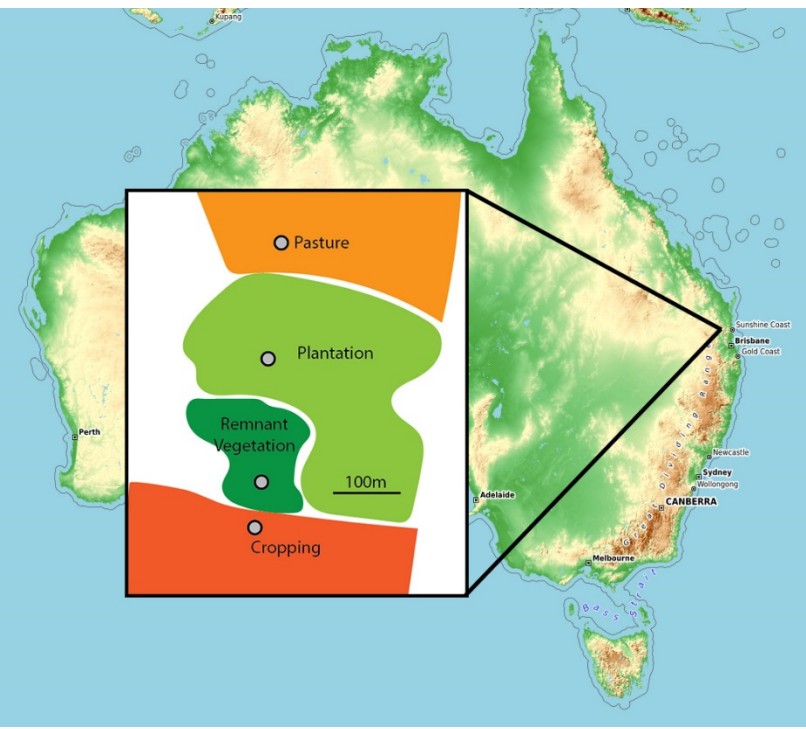

**Fig. 1: Schematic diagram of sampling design with sample locations and land use pattern illustrated. Getlost Maps.**


## 2.2 Bulk analyses

All samples were air-dried (~40 °C) and sieved to < 2 mm before further analyses were conducted. The electrical conductivity (EC) and pH were measured using a 1:5 soil/water suspension at 25 °C as described in Rayment and Lyons (2011). The total organic C and N were determined through Dumas high-temperature combustion with no soil pre-treatment (Rayment and
Lyons, 2011) since no carbonate-C was present (soil pH ≤ 5.5). Each sample was finely ground (< 0.5 mm) using a ceramic mortar and pestle and 1.00 g was weighed into combustion boats. The concentrations of total organic C and N were measured using a CN analyser (LECO Trumac CN analyzer, MI).

Exchangeable cations ($Al^{3+}$, $Ca^{2+}$, $Mg^{2+}$, $Na^+$, $K^+$) were determined by extraction in 1 M $NH_4Cl$ at pH 7 without the pre-
treatment for soluble salts as outlined in method 15A1 in Rayment and Lyons (2011). The extracts were then analysed using inductively coupled plasma optical emission spectroscopy (ICP-OES). The effective cation exchange capacity (ECEC) was determined by summating the exchangeable cations. The potentially plant-available P concentration was determined by extraction with bicarbonate (Colwell-P) and analysis in a SEAL AQ400 discrete analyser as described in method 9B1 of Rayment and Lyons (2011).


## 2.3 Microbial respiration

The susceptibility of the SOC to microbial decomposition was determined by measuring microbial respiration (by $CO_2$ release) in a 15-d incubation experiment by a method adapted from Niaz et al. (2022). There was a total of 20 experimental units, comprising five replicates for each of the four land uses. For each of the experimental units, 25 g of field soil was added to
250 mL jars and then wet to 80% field capacity. The soil water content at field capacity (-0.1 bar) was determined by wetting soil to saturation and then allowing it to drain for 24 h. To maintain 80 % field capacity throughout the incubation period, the jars were stored (with lids removed) in a humid environment at 25 °C and additional deionised water was added to the jars throughout the incubation experiment to keep a consistent weight representing 80% field capacity (±0.2 mL). Four blanks were

included to measure the background $CO_2$. The jars were covered with lids having two holes that had Luer locks inserted into these holes and arranged in a completely randomised design. The jar lids were closed (not allowing any air transfer) for 2 h prior to measurement. After 2 h, each jar was connected via rubber tubing to a $CO_2$ analyser (WMA-4 $CO_2$ analyser, John Morris USA) to record the concentration of $CO_2$. The first 3 d were allowed for equilibration, with data recorded every day thereafter for the first 5 d, and then at 2 d interval after this. The measurements were converted to mg $CO_2$-C $kg^{-1}$ soil $d^{-1}$ from the volume of $CO_2$ in the headspace of the respiration jars. Cumulative $CO_2$ produced during the incubation experiment was calculated by summation of the recorded $CO_2$ over each day for the incubation period.

## 2.4 Density fractionation

The density fractionation method was adapted from Kölbl and Kögel-Knabner (2004) and Steffens et al. (2009) (Supplementary Fig. S2). Briefly, 20 g bulk soil was added to a 200 mL sodium polytungstate (1.8 g $cm^{-3}$) solution and the floating free particulate organic matter (POM) ("fPOM") was collected by aspiration. The soil was not shaken within the SPT to minimise any disturbance. To separate the aggregate-occluded POM ("oPOM") from the remaining sample, the soil was subjected to sonification in two waves of 200 J $mL^{-1}$ (total of 400 J $mL^{-1}$) and then centrifuged. The use of 400 J $mL^{-1}$ for sonication was based upon a preliminary experiment to maximise recovery of POM at the lowest possible energy input, with this value being in accordance with previous studies for reactive soils (Asano and Wagai, 2014; Spielvogel et al., 2007). The floating oPOM was collected by aspiration. The fPOM was manually washed, and the oPOM was washed using pressure filtration (operated at 5 bar), with ultrapure Milli-Q water until the filtrate for both fractions was EC $< 5$ µS $cm^{-1}$. The remaining mineral-associated organic matter (MAOM) fraction was washed using pressure filtration to an EC $< 50$ µS $cm^{-1}$ and then sieved under gravity using a 53 µm steel sieve to separate the fine fraction-MAOM ($> 53$ µm) from the coarse fraction-MAOM ($< 53$ µm). The fPOM and oPOM were freeze dried, and the MAOM fractions were oven dried at 50 °C. The samples were finely ground for total C and N analysis by isotope ratio mass spectrometry (IRMS). The results for the C contents of each fraction were termed the 'fPOC' (free particulate organic C), 'oPOC' (aggregate-occluded organic C), 'coarse fraction-MAOC' (organic C associated with the coarse mineral fraction [$> 53$ µm]) and the fine fraction-MAOC (organic C associated with the fine mineral fraction [$< 53$ µm]). Similarly for N, the results giving the N contents of each fraction were termed the 'fPON', 'oPON', 'coarse fraction-MAON' and 'fine fraction-MAON'.

## 2.5 Synchrotron-based near edge X-ray absorption fine structure spectroscopy

All samples were air-dried ($\sim 40$ °C), sieved to $< 2$ mm and ground to a fine powder using a mortar and pestle. One composite sample per land use was generated by combining each of the five replicates. The composites were attached to a plate of stainless-steel using double sided C tape. Samples were analysed at the SXR beamline at the Australian Synchrotron (Clayton, Australia). The partial electron yield method was used to collect near edge X-ray absorption fine structure (NEXAFS) spectroscopy C K-edge spectra. To counter charging effects of soil minerals, a flood gun was used with partial electron yield. A calculated energy resolution of 0.05 eV at a 280 eV photon energy was obtained by setting the beamline grating exit slits to 20 µm. The spectra were acquired at an angle of 55° to the beam and over a photon energy range of 275 to 325 eV, at a 0.1 eV step size. For baseline correction, the pre-edge energy range was 270 and 275 eV and post-edge energy range was 325 and 340 eV. The energy of the beamline (270-340 eV) was calibrated by using a graphite standard. Concurrently, the sample NEXAFS and incident intensity ($I_0$) spectra were collected (Weng et al., 2017).

A photodiode measurement obtained in the UHV analytical chamber, and the $I_0$, were used to normalise the spectra. The double normalization method, as described by Stohr (2013), was utilised to carry out the normalisation. Data was normalised using

the Igor software (v8.04.2) with a pre- and post-edge linear subtraction. Double normalized spectra were deconvoluted using Athena 0.9.26 (Ravel and Newville, 2005). Deconvolution and curve fitting was conducted as described by Prietzel et al. (2018) using the peakfit procedure in Athena (Supplementary Fig. S3). An arctan function (energy: 290 eV, fixed height of 1) was used to represent the edge step. Six Gaussian peaks were then fitted for the six major C electron transition groups: (1) 284.7 eV, (2) 285.5 eV, (3) 287.3 eV, (4) 288.2 eV, (5) 289.0 eV, and (6) 289.8 eV with fixed widths of 0.4 eV. Two additional

Gaussian peaks were set at 292 and 294 eV (width 1.5 eV), representing broad additional 1s ➔ σ* transitions of saturated single covalent bonds and direct inner-shell ionisation (Prietzel et al., 2018). The contributions of the various C functional groups were calculated as the area under the Gaussian peaks divided by the sum of these areas.

### 2.6 Synchrotron-based infrared microspectroscopy

All samples were air-dried (~40°C) and sieved to < 2 mm. One composite sample per land use was generated by combining each of the five replicates. Typical free microaggregates (ca. 20-30) of 53-250 μm were humidified and then frozen at -20 °C. A diamond knife was used to cryo-ultramicrotome semi-thin sections (ca. 200 nm thickness) of microaggregates without embedding media. Thin soil sections were transferred to $CaF_2$ windows (13 mm diameter, 0.5 mm thickness). The samples were analysed at the Australian Synchrotron using the IRM beamline in transmission mode. A detection aperture to sample an

area of 5 μm × 5 μm was selected. 32 coadded scans (4 $cm^{-1}$ resolution) were taken to produce the maps (2.5 μm step size over ca. 50 μm × 50 μm).

The software OPUS 8.7.31 (Bruker Optik GmbH, Germany) was used to process spectral maps. Map profiles were created for specific absorbances which coincide with the peaks of relevant C functional groups, and O-H groups of clay minerals. A map

profile for absorbance at 3630 $cm^{-1}$ was created as this peak corresponds with the stretching vibrations of O-H clay minerals. Similarly, the peak at 1035 $cm^{-1}$ corresponds to the C-O stretching vibration of polysaccharide C, the C=C stretching of aromatic C (or N-H deformations) corresponds to a peak at 1600 $cm^{-1}$, and the peak at 2920 $cm^{-1}$ corresponds to the C-H stretching of aliphatic biopolymers. On either side of the absorbance peaks [3550–3740 $cm^{-1}$ (O–H groups of clays), 950–1170 $cm^{-1}$ (polysaccharide C), 1500–1750 $cm^{-1}$ (aromatic C), 2800–3000 $cm^{-1}$ (aliphatic C)] appropriate baseline points were

selected so that the integrated area under each absorbance peak could be applied to the map.

### 2.7 Statistical analyses

Repeated measures analyses were performed for all data using a linear mixed model framework (REML) using the Genstat software (Version 23.1.0.651). Each analysis for organic C content, total organic C per fraction, deconvolution of NEXAFS

consisted of a fixed model of land use, fraction, and their associated interactions, and a random effect of replicate (Fang et al., 2022). Tukey's post hoc testing was used to identify significant differences based on the REML mixed effects model ($p < 0.05$). . To determine the significance level, a one-way analysis of variance (ANOVA) was completed with land use as a factor, for the bulk soil properties, respiration rate and cumulative respiration at each time point to determine if there were any significant differences between the sites using R version 4.2.1 (R Core Team, 2022). The ANOVA was checked for the normal

distribution of residuals and homogeneity of variances (Supplementary Fig. S4). Where the ANOVA identified a significant difference in land use ($p < 0.05$), the least significant difference (LSD) (95 % confidence level for each comparison) was calculated to determine which land uses differed significantly from one another. A two-way ANOVA was completed with fraction and land use as factors for the density fractionation data. The data frame was then split into each fraction, where one-way ANOVA was used to identify significant differences between land uses followed by calculating the LSD to compare the

means of each land use.

For the synchrotron-based IRM, correlations between the polysaccharide C, aromatic C or aliphatic C and the amount of clay were examined using simple linear regression. The relative strength of associations was indicated by the regression gradients, and the residual variability affiliated with the associations was indicated by the $R^2$ coefficients. It is important to note that both the synchrotron-based IRM and NEXAFS techniques, as performed in the present study, are considered qualitative methods for characterising C forms and distribution.

## 3 Results

### 3.1 Bulk C contents and pools

There was a significant difference in the pH (1:5 suspension) between the various land uses ($p = 0.026$), with the plantation soil having a significantly lower pH (5.0) compared with the pasture soil (5.5, Table 1). For EC (1:5 suspension), no significant differences were found between the land uses, with an average value of 0.11 dS m$^{-1}$ (Table 1). There were substantial differences in the ECEC values ($p < 0.01$), which was significantly lower for the cropped soil (3.0 cmol$_c$ kg$^{-1}$) than the other land uses (7.5 cmol$_c$ kg$^{-1}$, Table 1). Furthermore, there were significant differences in the Colwell-P concentration ($p < 0.01$) with the crop (58 mg kg$^{-1}$) and pasture soils (51 mg kg$^{-1}$) both having higher extractable P than the remnant vegetation (18 mg kg$^{-1}$) and plantation (19 mg kg$^{-1}$, Table 1).

**Table 1: Mean values (standard errors in parentheses) of the bulk soil from each land use. Lower-case letters reflect least significant differences ($P < 0.05$) for pH, electrical conductivity (EC), effective cation exchange capacity (ECEC), phosphorus and total organic carbon and total N. Values are the average of five replicates.**

| Land Use | pH (1:5 water) | EC (dS m$^{-1}$) (1:5 water) | ECEC (cmol$_{(+)}$ kg$^{-1}$) | Colwell P (mg kg$^{-1}$) | Total organic C (g kg$^{-1}$ soil) | Total N (g kg$^{-1}$ soil) | C:N ratio |
|---|---|---|---|---|---|---|---|
| **Remnant Vegetation** | 5.2$^{ab}$ (0.13) | 0.13 (0.19) | 8.7 (0.80)$^a$ | 18$^b$ (1.2) | 81$^a$ (13) | 5.9$^a$ (0.74) | 14 (0.50) |
| **Pasture** | 5.5$^a$ (0.09) | 0.081 (0.01) | 6.3 (0.53)$^b$ | 51$^a$ (6.5) | 38$^b$ (2.0) | 3.1$^b$ (0.08) | 12 (0.41) |
| **Plantation** | 5.0$^b$ (0.07) | 0.12 (0.02) | 7.5 (0.87)$^{ab}$ | 19$^b$ (3.7) | 63$^a$ (8.6) | 3.9$^b$ (0.46) | 16 (0.33) |
| **Cropped** | 5.2$^{ab}$ (0.09) | 0.10 (0.014) | 3.0 (0.29)$^c$ | 58$^a$ (6.0) | 18$^b$ (1.9) | 1.6$^c$ (0.13) | 12 (0.40) |

Notably, the total SOC content differed between land uses ($p < 0.01$) and decreased in the order of remnant vegetation (81 g kg$^{-1}$) ≈ plantation (63 g kg$^{-1}$) > pasture (38 g kg$^{-1}$) ≈ cropped (18 g kg$^{-1}$) (Table 1). Hence, the conversion of remnant vegetation to cropped resulted in a 78% decrease in SOC in the surface 0-10 cm. Changes in total N were similar to those described above for C ($p < 0.01$), being highest in the remnant vegetation soil (5.9 g kg$^{-1}$) and lowest in the cropped soil (1.6 g kg$^{-1}$). The C:N ratio varied from 12 in the cropped soil to 14 in the remnant vegetation, 12 in the pasture and 16 in the plantation (Table 1).

### 3.2 Microbial respiration

The cropped soil had a significantly lower microbial respiration rate (0.0069 mg $CO_2$-C g$^{-1}$ soil day$^{-1}$) compared to the pasture, remnant vegetation and plantation soils (0.020 mg $CO_2$-C g-soil$^{-1}$ day$^{-1}$, Fig. 2a). When examined as cumulative respiration

over the 15-d experimental period, the cropped soil had a significantly lower cumulative respiration (0.13 mg $CO_2$-C g$^{-1}$ soil) compared to all other land uses by the final day of incubation (0.29 mg $CO_2$-C g$^{-1}$ soil; Fig. 2b).

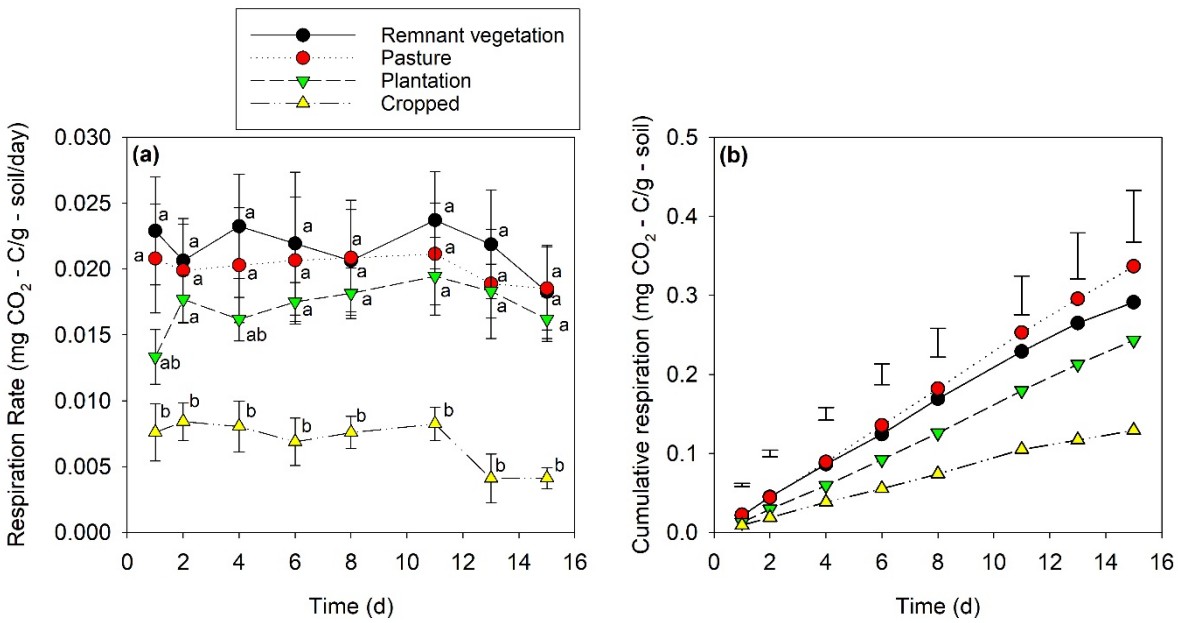

220   **Fig. 2: (a) Mean (with standard error) respiration rates of soil for four land uses. Letters indicate the least significant difference (*P* <0.05) between the land uses at each time. (b) Cumulative respiration for four land uses across a 15-d incubation period. Each point represents the mean of five replicates. Error bars indicate least significant differences (*P* <0.05) between land uses at each time.**

### 3.3 Density fractionation

225   In the remnant vegetation soil, the fPOC fraction (22.3 mg C g$^{-1}$ bulk soil) and fine fraction-MAOC (19.8 mg C g$^{-1}$ bulk soil) were the two largest fractions, with the coarse fraction-MAOC (7.59 mg C g$^{-1}$ bulk soil) and the oPOC (9.29 mg g$^{-1}$ bulk soil) being considerably smaller (Fig. 3a). Indeed, for the remnant vegetation, the fPOC and the fine fraction-MAOC accounted for 71% of the total bulk organic C (Fig. 3b). Similarly for N, the fine fraction-MAON (2.14 mg N g$^{-1}$ bulk soil) and fPON (1.58 mg N g$^{-1}$ bulk soil) were the largest pools for the remnant vegetation soil (Supplementary Fig. S5).

230

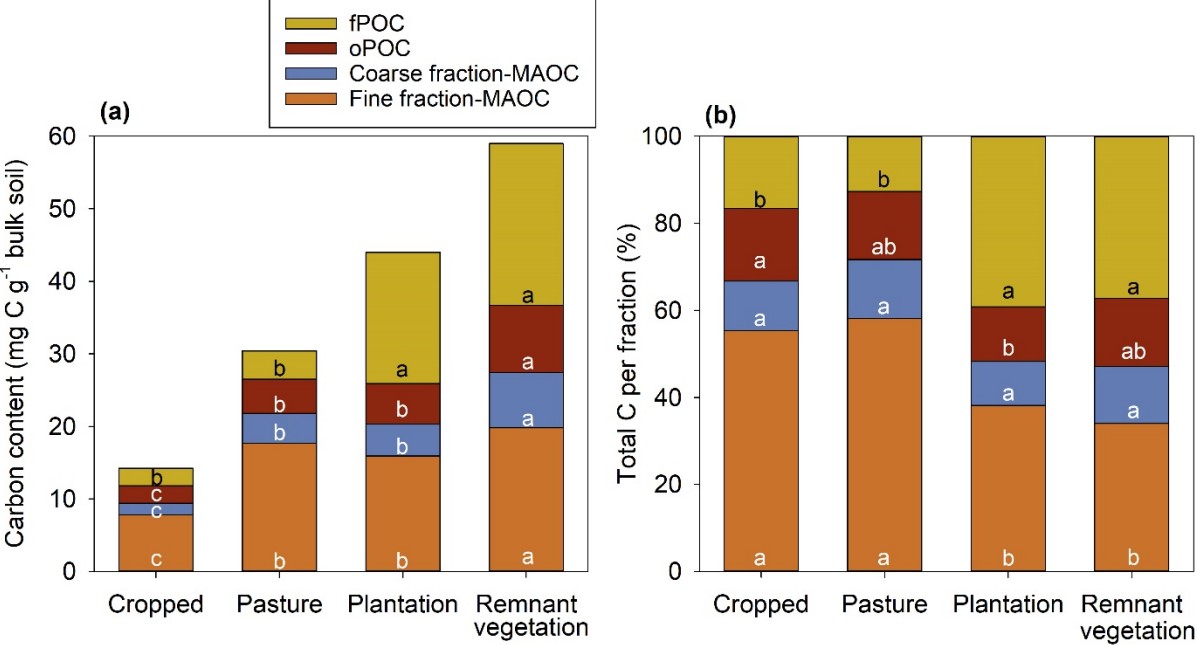

**Fig. 3: Organic carbon content (a) and the total organic C per fraction (b) for each soil fraction within the bulk soil from topsoils (0-10cm) collected from four land uses. fPOC is free particulate organic carbon, oPOC is aggregate-occluded particulate organic matter, fine fraction-MAOC is coarse grained (> 53 μm) mineral-associated organic carbon and coarse fraction-MAOC is fine grained (> 53 μm) mineral-associated organic carbon. Lower-case letters indicate least significant differences (*P* < 0.05) between the same fractions across land uses according to REML mixed effects model and Tukey's post hoc testing.**

The decrease in total organic C observed for all three other land uses compared to the remnant vegetation soil (Table 1) was associated with a decrease in the absolute contribution of all four fractions for the cropped soil and pasture (fPOC, oPOC, the coarse fraction-MAOC and fine fraction-MAOC) and all fractions except the fPOC for the plantation (Fig. 3a). However, the relative contribution of each fraction to the total organic C varied with land use change. For the conversion of remnant vegetation to cropped, the fPOC fraction was the most labile (least stable) with land use change, with its relative contribution to the total SOC decreasing from 37% in the remnant vegetation soil to only 17% in the cropped soil (Fig. 3b). In contrast, the fine fraction-MAOC was the most stable fraction, with its relative contribution to total SOC increasing from 37% in the remnant vegetation soil to 55% in the cropped soil. In addition, the loss in fPON content was even greater than observed for fPOC with a decrease from 1.6 mg N g$^{-1}$ bulk soil in the remnant vegetation to only 0.12 mg N g$^{-1}$ bulk soil in the cropped soil, a decrease of 92% (Supplementary Fig. S5), with cropped soil also having a higher C:N ratio (20.5) than remnant vegetation (14.1) (Supplementary Table S2).

Restoration of cropped land to pasture significantly increased the absolute C content of all fractions except for the fPOC (Fig. 3a). However, these increases in the oPOC and MAOC fractions were not enough to restore SOC to the original values observed under remnant vegetation (Fig. 3a). The fine fraction-MAOC pool was still the largest fraction for the pasture soil, accounting for 58% of the total C, with a similar value observed for the cropped soil (55%) (Fig. 3b). A similar pattern was observed for organic N in the various fractions as described for organic C (Supplementary Fig. S5).

Finally, restoration of cropped soil to plantation resulted in a significant increase in both organic C and N content of all fractions (Fig. 3a, Supplementary Fig. S5). However, despite the conversion of cropping to pasture increasing C and N, contents still remained lower for all fractions compared to the remnant vegetation, with the exception being for the fPOC fraction for the plantation soil (18.1 mg C g$^{-1}$ bulk soil) which was not significantly different from the remnant vegetation soil (22.3 mg C g$^{-1}$ bulk soil).

## 3.4 Land use impacts on C forms: Synchrotron-based near edge X-ray absorption fine structure spectroscopy

To examine the effect of long-term land use change on the forms of organic C and N within the soil, synchrotron-based NEXAFS was used. The C K-edge spectra had clear peaks for all land uses at 284.7 eV indicating the presence of quinones (C=O), 285.5 eV indicating aromatic C/double-bonded alkyl C (C=C), 287.3 eV indicating aliphatic C/phenolic C-OH, (aromatic C with side chain N-substituted aromatic C), 288.2 eV indicating aliphatic C (alkyl C, C-H), 289.0 eV indicating carboxyl C-OOH functional groups, and 289.8 eV indicating O-alkyl-C (C-OH)  (Fig. 4a). The number of organic C forms observed using NEXAFS did not change with land uses. Importantly, however, we observed differences in the intensities of these peaks within various fractions. In remnant vegetation, the most dominant C function group for all fractions was carboxylic-C, ranging from 42-44% in the POM to 47-48% in the MAOM (Table 2). This was followed by O-alkyl-C for all fractions, accounting for 26% across all fracitons. Aliphatic-C was the next most dominant C functional group for fine MAOC accounted for 18%. This pattern was observed for the rest of the land uses, namely, dominated by carboxylic-C followed by O-alkyl-C and aliphatic-C. The proportion of C functional groups in the fine MAOC was similar across all land uses with 20% for aliphatic-C, 47% for carboxylic-C, and 26% for O-alkyl-C.

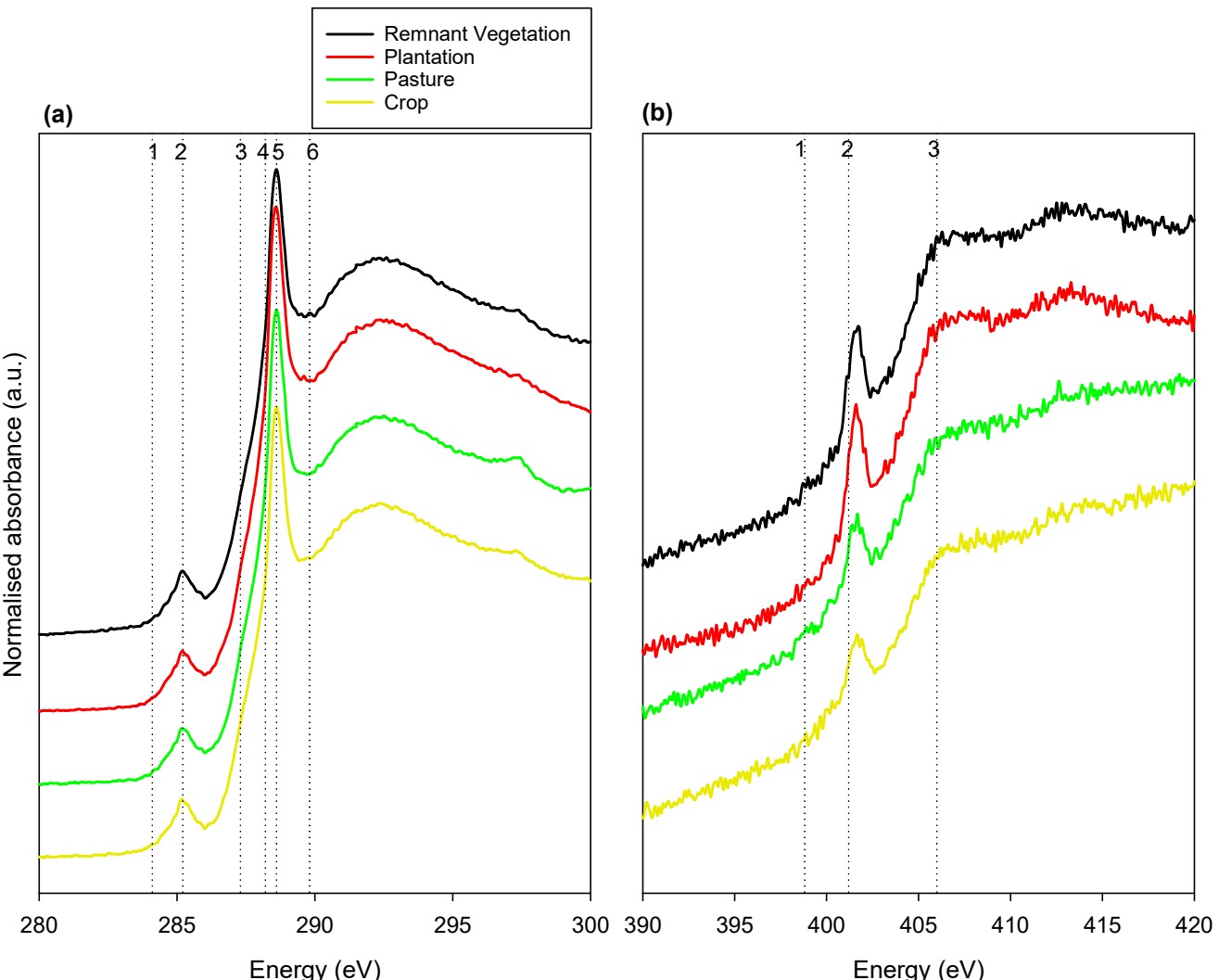

**Fig. 4: Double normalised (a) C K-edge spectra and (b) N K-edge spectra of finely ground bulk soil samples from each land use produced by near edge X-ray absorption fine structure. Spectra have been stretched in the y direction to visualise each land use. (a) Vertical lines indicate (1) quinones at 284.7 eV, (2) aromatic C at 285.5 eV, (3) phenolic C-OH at 287.3 eV, (4) aliphatic C at 288.2 eV, (5) carboxyl C-OOH at 289.0 eV and (6) O-alkyl-C at 289.8 eV. (b) Vertical lines indicate (1) aromatic N in 6-membered rings at 398.8 eV, (2) amide at 401.2 eV, and (3) alkyl-N at 406 eV.**

**Table 2: Relative proportion of organic C functional groups in the various fractions under different land uses identified by C (1s) near edge X-ray absorption fine structure (NEXAFS) spectroscopy. fPOC is free particulate organic carbon, oPOC is aggregate-occluded particulate organic matter, coarse fraction-MAOC is coarse grained (> 53 µm) mineral-associated organic carbon and fine fraction-MAOC is fine grained (< 53 µm) mineral-associated organic carbon. Quinones (C=O): 284.7 eV, aromatic C: 285.52 eV, double-bonded alkyl C (C=C); phenolic C-OH: 287.3 eV, aromatic C with side chain N-substituted aromatic C; aliphatic C: 288.2 eV, alkyl C, C-H; carboxyl C-OOH functional groups: 289.08 eV; O-alkyl-C: 289.8 eV, C-OH. Statistical significance according to REML mixed effects model based on Tukey's post hoc testing ($p$ values.)**

| Fractions | Land use | Quinine | Aromatic | Phenolic | Aliphatic | Carboxylic | O-alkyl-C |
|---|---|---|---|---|---|---|---|
| | | | | | (%) | | |
| fPOC | Cropped | 0.67±0.17 | 5.3±0.27 | 4.8±0.46 | 21±0.27 | 41±0.87 | 26±0.25 |
| | Pasture | 0.79±0.31 | 5.3±0.41 | 4.2±0.35 | 22±0.23 | 41±0.68 | 26±0.36 |
| | Plantation | 0.29±0.07 | 5.9±0.79 | 3.3±0.56 | 25±0.43 | 40±1.8 | 26±0.15 |
| | Remnant vegetation | 0.06±0.02 | 5.3±0.07 | 5.3±0.28 | 20±0.13 | 44±0.36 | 26±0.13 |
| oPOC | Cropped | 0.10±0.03 | 5.8±0.13 | 4.4±0.13 | 19±0.17 | 42±0.21 | 28±0.20 |
| | Pasture | 0.24±0.07 | 6.2±0.33 | 4.3±0.35 | 21±0.32 | 41±0.70 | 27±0.24 |
| | Plantation | 0.17±0.06 | 6.4±0.63 | 3.9±0.43 | 23±0.22 | 40±1.0 | 27±0.19 |
| | Remnant vegetation | 0.07±0.03 | 5.9±0.53 | 5.3±0.61 | 19±0.14 | 42±1.2 | 27±0.14 |
| Coarse MAOC | Cropped | 0.09±0.04 | 3.9±0.37 | 2.7±0.36 | 20±0.16 | 47±0.78 | 26±0.07 |
| | Pasture | 0.11±0.03 | 4.1±0.51 | 2.9±0.53 | 19±0.12 | 48±0.94 | 26±0.11 |
| | Plantation | 0.07±0.02 | 4.1±0.46 | 2.8±0.47 | 20±0.18 | 48±1.0 | 26±0.13 |
| | Remnant vegetation | 0.14±0.02 | 3.4±0.46 | 2.4±0.52 | 20±0.07 | 48±0.96 | 26±0.12 |
| Fine MAOC | Cropped | 0.08±0.03 | 3.7±0.32 | 2.5±0.28 | 19±0.03 | 48±0.52 | 26±0.07 |
| | Pasture | 0.18±0.04 | 4.7±0.71 | 3.1±0.61 | 20±0.55 | 47±1.2 | 25±0.47 |
| | Plantation | 0.19±0.03 | 3.2±0.31 | 1.8±0.30 | 20±0.41 | 49±0.54 | 26±0.33 |
| | Remnant vegetation | 0.12±0.04 | 4.7±0.31 | 3.6±0.26 | 18±0.38 | 47±0.74 | 26±0.14 |
| Statistical significance | fPOC | 0.057 | 0.777 | 0.044 | <0.001 | 0.116 | 0.065 |
| | oPOC | 0.147 | 0.798 | 0.081 | <0.001 | 0.211 | 0.002 |
| | Coarse MAOC | 0.242 | 0.731 | 0.905 | <0.001 | 0.884 | 0.451 |
| | Fine MAOC | 0.19 | 0.159 | 0.067 | 0.061 | 0.341 | 0.105 |

In addition, the N K-edge spectra showed defined peaks at 398.8 eV indicating aromatic N in 6-membered rings, 401.2 eV indicating amides, and 406 eV indicating alkyl-N functional groups (Fig. 4b). The amide peak was weaker in the cropped soil compared with the other land uses. However, the aromatic N 6-membered rings and alkyl-N peaks were similar between all land uses.

**3.5 Distribution of C forms: Synchrotron-based IRM**

Synchrotron-based IRM analyses were used to investigate the two-dimensional distribution of C forms within the soil on a micro-scale, with spectral maps showing the forms of C (aliphatic C, aromatic C, and polysaccharide C) and mineral-OH within 200 nm semi-thin sections taken from intact microaggregates (<250 μm) (Fig. 5).

For the soil from remnant vegetation (Fig. 5a), it was found that the lateral distribution of clay minerals within the microaggregates correlated strongly with the distribution of aliphatic C ($R^2 = 0.96$), aromatic C ($R^2 = 0.84$) and polysaccharide C ($R^2 = 0.78$). It was observed that aliphatic C in the pasture soil was also correlated with clay minerals ($R^2 = 0.58$) although correlations between aromatic C and clay minerals, as well as between polysaccharide C with clay minerals, were weaker in this soil (Fig. 5b). Similarly, for the plantation soil, there were strong correlations between clay minerals and both polysaccharide C and aliphatic C, whilst aromatic C was only weakly correlated with clay minerals (Fig. 5c). In a similar manner, for the cropped soil (Fig. 5d), strong correlations were found between the distribution of clay minerals and both aliphatic C ($R^2 = 0.92$) and aromatic C ($R^2 = 0.73$) but there was no clear correlation between the distribution of polysaccharide C and clay minerals ($R^2 = 0.24$). The gradients for the regressions of aliphatic C and aromatic C with clay minerals were only marginally lower in the cropped soil than the remnant vegetation. Additionally, for polysaccharide C, the degree of association with clay minerals and the gradient decreased in the order: remnant vegetation soil > plantation soil > pasture soil > cropped soil.

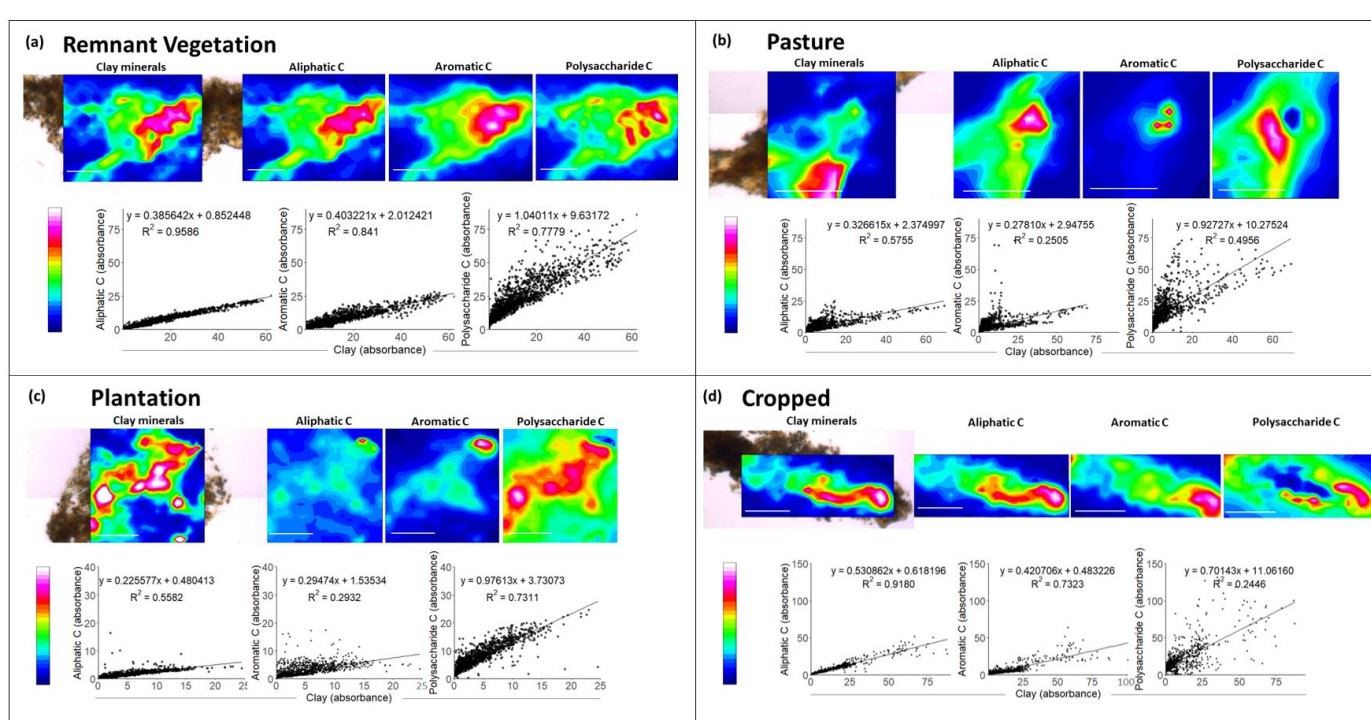

**Fig. 5: Spectral maps showing the distribution of mineral-OH, aliphatic C, aromatic C and polysaccharide C obtained through synchrotron-based infrared microspectroscopy by analysis of sections (200 nm thickness) obtained from intact microaggregates of soil from four land uses (remnant vegetation, pasture, plantation and cropped) (bars are 25 μm). The spectral map of one of the two to three replicates analysed for each land use is shown. Maps were obtained from 32 coadded scans (4 cm$^{-1}$ resolution) with a 2.5 μm step size. The spectra obtained from each map were used to form regression analyses. Pixels where the absorption peak could not be detected above the baseline noise were excluded from regressions.**

# 4 Discussion

### 325 4.1 Land use change to cropping profoundly decreases SOC and N contents but partial protection is provided by occlusion or binding to the mineral fraction

In this semi-arid subtropical Ferralsol, conversion of remnant vegetation to long-term cropping resulted in the loss of 78% of total SOC at a depth of 0-10 cm after 72 y (Table 1). Globally, long-term cropping has been identified to reduce SOC stocks by 20-60% (Kopittke et al., 2017). The high loss of C in the cropped soil in the present study could possibly be attributed to
the warm, subtropical climate compared to other studies conducted on soils of temperate regions. In addition, Ferralsols have been found to lose a greater proportion of surface-layer SOC than other soil types (Hartemink, 1997) which may explain the very high C loss observed in the present cropped soil. In addition to changes in SOC contents caused by land use conversion to cropped land, it was observed that there was a 74% decrease in total N, with decreases also reported in previous studies under long-term cropping (Dalal et al., 2021b; Kopittke et al., 2017).


Not only did land use change cause marked changes in total SOC and N contents, but it was also found that there were substantial changes in the various SOC and N fractions within the soil. Overall, it was observed that the fine fraction-MAOM made the largest contribution to the total SOC and N contents regardless of land use (Fig. 3a and Supplementary Fig. S5). These findings regarding the importance of the fine fraction-MAOC are consistent with previous studies, even following land
management changes (Degryze et al., 2004; John et al., 2005; Kleber et al., 2015). In addition, MAOM has been identified to be one of the largest N pools and total N has been shown to be highly correlated with C for the MAOM fraction for a variety of soils (Kirkby et al., 2011).

Although the fine fraction-MAOC was the dominant fraction, it was found that conversion of remnant vegetation to cropped
land caused the loss of C and N from all fractions (i.e., decreased their absolute size, Fig. 3a and Supplementary Fig. S5), with this being consistent with previous studies for both organic C (Degryze et al., 2004; John et al., 2005; Six et al., 1998) and N (Yang et al., 2022). Although the size of all fractions decreased for both organic C and N, it was noted that the magnitude of the decrease was greatest for the fPOC fraction and least for the fine fraction-MAOC (Fig. 3a). Indeed, the fPOC fraction accounted for 37% of the total SOC in the remnant vegetation but only 17% in the cropped soil, while the fine fraction-MAOC
accounted for 34% of the total SOC in the remnant vegetation but 55% in the cropped soil (Fig. 3b). This observation that the fPOC fraction is the most susceptible to loss upon conversion to long-term cropping for a subtropical soil is consistent with previous studies from temperate (Six et al., 1998; Besnard et al., 1996) and tropical regions (Ashagrie et al., 2007). This marked loss of fPOM upon land use change to long-term cropping is critical, given that this is the most labile fraction and hence plays an integral role in nutrient cycling and soil biological properties due to its nature as an easily accessible substrate for
microorganisms.

It is also noteworthy that although fPOM was the most labile fraction, the occlusion of the POM within soil aggregates (i.e., oPOM) was important for its resilience – whilst the relative contribution of fPOM significantly decreased from 37 to 17% upon conversion to long-term cropping, oPOM remained relatively constant (15.6 to 16.7%, Fig. 3b). Thus, these data highlight
not only the importance of the binding of SOC to the mineral fraction (MAOC), but also the importance of soil structure to enable the occlusion of POM (oPOM) in increasing resilience and stability of SOC in subtropical, semi-arid soils.

### 4.2 The marked loss of labile SOC resulted in decreased microbial activity

The loss of 78% of the SOC in this subtropical soil at 0-10 cm depth upon conversion to long-term cropping (Table 1), coupled
with the preferential loss of the fPOM fraction (Fig. 3a), resulted in a significant decrease in microbial functioning as measured

using microbial respiration rate. Indeed, it was observed that respiration was significantly lower in the cropped soil than for all other land uses (Fig. 2a). Reduced respiration in surface soils has been shown to be associated with lower SOC levels given that SOC is the substrate for soil microorganisms (Fan et al., 2015; Wang et al., 2013). Thus, the loss of 78% of the SOC upon conversion to cropped land not only increases $CO_2$ emissions to the atmosphere (as a greenhouse gas), the loss of SOC also adversely impacts upon the functioning of soil as shown through decreases in microbial respiration.

It must also be noted that it cannot be excluded that the decrease observed in microbial respiration rate in the cropping system could potentially, to come extent, be associated with the application of pesticides. In this regard, pesticides have been reported to adversely effect the soil microbial community, although the there are conflicting reports in this regard and their effects remain unclear (Hussain et al., 2009).

### 4.3 Changing land use from cropping back to pasture or plantation can partially restore SOC and microbial activity

Whilst conversion of remnant vegetation to cropped land resulted in the loss of 78% of the SOC, this could be reversed, although not fully, by conversion of cropped soil to pasture or plantation. Indeed, although not significantly different, conversion of the cropped land to pasture increased the total SOC content by 111 %, whilst conversion of cropped to pasture and then to plantation increased SOC by 250 % (Table 1). Importantly, SOC remained lower in the pasture soil (38 g kg$^{-1}$) than in the remnant vegetation (81 g kg$^{-1}$), but in the plantation soil (63 g kg$^{-1}$), the SOC was increased to a level that was not significantly lower than the remnant vegetation (Table 1). Furthermore, conversion of the cropped soil to pasture significantly increased the total N content by 94 % (Table 1).

This increase in both organic C and N when cropped soil was converted to pasture was associated with increases in all fractions with the exception of fPOM (i.e., increases in oPOM, fine fraction-MAOM, coarse fraction-MAOM) (Fig. 3a, Supplementary Fig. S5). Of these, the largest absolute increase was for the fine fraction-MAOC, which increased from 7.8 mg g$^{-1}$ bulk soil in the cropped soil to 18 mg g$^{-1}$ bulk soil in the pasture. This increase in organic C and N content of the fine fraction-MAOM upon conversion of cropped land to pasture in the present study is likely due to the higher below- and above-ground input of soil organic matter (SOM) in grasslands, and lower intensity disturbance in grasslands. In addition, Ferralsols are rich in iron-oxides which can act as an integral regulator for MAOM formation and can lead to rapid accrual of MAOC (Kleber et al., 2015; Ye et al., 2019). This observed pattern of restoration of the SOC pools upon conversion to pasture is interesting in the context of the soil continuum model proposed by Lehmann and Kleber (2015). It is thought that the increase in biomass input that occurs with the transition from cropping to pasture would first regenerate the fPOM pool, but presumably this pool continued to be rapidly converted to oPOM and ultimately mineral-associated forms. Perhaps this fPOM pool has not yet been restored in the pasture soil as a result of the relatively low rate of biomass production in this environment or the fast turnover of POM within this subtropical climate.

Not only did conversion from cropped land to pasture generally increase SOC in the various fractions, but conversion to plantation also resulted in a significant increase in the C and N content of all fractions (Fig. 3a, Supplementary Fig. S5). Interestingly, the largest increase in organic C and N for the plantation soil was observed for the fPOM fraction, with fPOC and fPON both increasing by more than 656% (Fig. 3a, Supplementary Fig. S5). Therefore, the restoration of the labile fPOM pool with conversion of cropped land to plantation was a key component of the increase in total SOC seen with this change (Table 1). This increase in the C content of the majority of fractions due to the conversion of cropped land to pasture and plantation, including a large increase in labile fPOC from conversion to plantation, resulted in an increase in soil respiration (Fig. 2b) and thus the restoration of microbial activity. However, although the plantation soil was able to restore the total C to

a level not significantly less than remnant vegetation (Table 1), the organic C content of the MAOC fractions and oPOC, still remained significantly lower in the plantation soil compared to the remnant vegetation (Fig. 3a).


This study illustrates the difficulty in restoring C as part of GHG removal (i.e., negative emission technologies, NETs) within semi-arid landscapes of subtropical regions, even in situations where productive land is removed from agriculture and returned to plantation. Indeed, even 39 y after converting cropped soil to pasture, SOC contents remained markedly lower in pasture (38 g kg$^{-1}$) than remnant vegetation (81 g kg$^{-1}$). Conversion of cropped land to plantation (63 g kg$^{-1}$) did restore total SOC

contents to a level not significantly less than the remnant vegetation, but most C fractions remained significantly lower. Given that large-scale deployment of NETs are required to keep warming below 1.5°C, this will represent a substantial challenge whilst also continuing to increase food production from soil (Kopittke et al., 2019).

### 4.4 Speciation and distribution of organic C associated with land use conversion

Despite the significant loss in total SOC, fPOC and MAOC with long-term cropping, there was no noticeable shift in C functional groups in bulk soils across any of the land uses (Fig. 4a). The C K-edge spectra indicated that quinones, aliphatic C/phenolic C-OH, aromatic C and carboxyl C-OOH were present within all land uses. Previous studies have concluded that bulk soils which differed in climate, vegetation compositions and mineralogy (Lehmann et al., 2008; Solomon et al., 2005) or organic matter at different stages of degradation (Solomon et al., 2007), have markedly similar C forms. This is confirmed by

estimating the proportion of C functional groups using deconvolution of NEXAFS peaks in the present study (Table 2). The proportion of C functional groups were similar for all SOC fractions across land uses indicating the SOC persistence was not a result of the presence of complex C structures rather a collection of simple and similar C functional composition.

For the N K -edge spectra, all land uses exhibited peaks likely corresponding to aromatic N in 6-membered rings, amide and

alkyl-N (Fig. 4b). However, the amide peak for the cropped soil was weaker than the other land uses which may suggest that this form of N is being more heavily consumed or mineralised by microbes due to increased soil disturbance in the cropped soil.

Given that there were no pronounced differences when examining bulk soil but notable shifts across various fractions, sections

taken from intact microaggregates were examined using synchrotron-based IRM. It was found that the distribution of C across microaggregate sections were highly heterogenous (Fig. 5) as has been observed in prior studies (Lehmann et al., 2008; Wan et al., 2007). It was noted that there was no clear gradient of SOC from the surface to the core of the microaggregates. This agrees with previous studies which examined the SOC distribution of microaggregates using similar methods (Hernandez-Soriano et al., 2018), and by using nanoscale secondary ion mass spectrometry (Kopittke et al., 2018; Steffens et al., 2017).

Thus, this evidence does not support the hypothesis that microaggregates are formed by mineral particles encapsulating organic fragments (Six et al., 1998; Tisdall and Oades, 1982).

It was also noteworthy that there were clear correlations between the various forms of C and clay minerals regardless of land use, which would suggest that mineral-associations are an integral mechanism which stabilises these C forms (Fig. 5). This

observation from *in situ* IRM analyses from the intact microaggregates supports observations from the fractionation analysis which demonstrated that the binding of C to the mineral fraction (i.e., the MAOC) is critical in determining SOC persistence, even in long-term cropped soil (Fig. 3). This study is the first to utilise *in situ* IRM analyses on the fine scale to confirm fractionation findings regarding the importance of MAOC in C persistence. This information is critical in developing

mechanistic models to enable better predictions of SOC persistence and the effects of land use change on SOC concentrations for climate change mitigation (Lehmann et al., 2020b).

Not only did the *in situ* IRM analyses confirm the importance of clay minerals in the binding and protection of SOC, but also differences were observed in the distributions of forms of C within the sections. More specifically, the correlation between aliphatic C and clay minerals was strong in both the remnant vegetation and the cropped soil. This indicates that aliphatic C remained strongly associated with clay minerals under cropping, which goes against past findings that suggest long-term cropping reduces mineral-associations with aliphatic C (Hernandez-Soriano et al., 2018). Considering that aliphatic C (wax layers on roots and leaves of terrestrial higher plants and metabolites and sugars from microbial debris, Jansen et al. (2010)) is selectively stabilised by iron oxides (Adhikari and Yang, 2015), which are abundant in Ferralsols, and that fine fraction-MAOC was proportionally the largest C pool for the cropped soil (Fig. 3b), it is likely that mineral-associated aliphatic C is one of the remaining persistent C forms in the cropped soil. This hypothesis is supported by the significant reduction in soil respiration for the cropped soil compared with the other land uses (Fig. 2). This suggests that there are few easily degradable forms of C remaining in the cropped soil and only persistent (not easily degraded) forms of C, such as persistent aliphatic C moieties, have remained in strong association with clay minerals.

Mineral-associations with polysaccharide C decreased in the order of remnant vegetation soil > plantation soil > pasture soil > cropped soil (Fig. 5). Polysaccharide C is often a more easily degradable C form than aliphatic C, and polysaccharide C was found to have a decreased degree of association with minerals with long-term cropping in several previous studies (Hernandez-Soriano et al., 2016; Solomon et al., 2007; Solomon et al., 2005). In addition, Solomon et al. (2005) found that polysaccharide C was reduced in the order of natural forests > plantations > cropping, as was observed here, likely due to accelerated mineralisation with cropping, as well as lower biomass input under cropping.

### 4.4 Study limitations

Given that these five samples collected at random points within a single treatment unit, they are considered to be pseudoreplicates (Hurlbert, 1984). No information was available for the sites and their properties from prior to land-use change (i.e. 72 y prior) and nor was any information available for the sites when they were converted from cropping to pasture or from pasture to plantation. Rather, the present study uses the space-for-time substitution approach as an alternative to long-term studies and has been regularly used for examining the effects of land disturbance (Pickett, 1989). One of the main issues associated with the space-for-time substitution approach is that there may be underlying spatial variability that is not accounted for and which can alter findings (Pickett, 1989). In this regard, the use of mixed-effects models allow us to compare the variance within the plots to the variance of the treatments with confidence.

We also note that it has been reported for subtropical soils of Queensland (Australia) that climate change is resulting in an overall slight increase in SOC over time (Dalal et al., 2021a), and although climate change would impact all land-uses in the present study, the effects on increased biomass production were likely more pronounced for the remnant vegetation treatment than for the other land-uses.

Finally, it is also noted that the cropped soils have had fertiliser applied whilst no fertilisers have been applied to the pasture or plantation soils (Supplementary Table S1). In this regard, the application of the low rates of fertiliser to the cropping soil would have impacted upon plant growth and the C and N in these soils. Indeed, it is known that application of fertilisers can alter concentrations of SOC, including in soils of Queensland (Australia) (Jha et al., 2022). Regardless, given that fertilisation

is part of typical management of these cropping soils and that the aim of this study was to examine the effect of land-use change and management (including cropping) on SOC, this is not considered to be a problem.

**5 Conclusions**

This study has shown that long-term use of a subtropical Ferralsol in Australia for cropping resulted in the loss of > 70% of both organic C and N compared with remnant vegetation. The organic C that was bound in the mineral fraction of the soil (i.e., the fine fraction-MAOC) was the most protected, whilst the fPOC fraction was the most susceptible to loss and had the lowest concentration in the cropped soil. Due to this loss of organic C, and especially the preferential loss of labile organic C, this research showed that microbial activity was greatly reduced in the cropped soil, with this having important implications for

soil fertility and functioning. Not only did fractionation demonstrate the importance of the binding of C to the mineral fraction in regulating C persistence, but *in situ* analyses also confirmed the strong binding of organic C to clay minerals, even in soil used for cropping for 72 y. Interestingly, despite the profound loss of organic C and N upon long-term cropping, NEXAFS analyses demonstrated that the forms of organic C within bulk soils were actually similar across all land uses. However, aliphatic-C was enriched in the fine MAOC fraction (< 53 µm) regardless of land uses indicating the role of microbial

processing in organo-mineral interactions. Furthermore, the enrichment of carboxylic C in the fPOC fraction in the cropped soil suggested the vulnerability of SOC to be decomposed after long-term cropping. Finally, it was also examined whether conversion of this subtropical Ferralsol used for cropping to either pasture or plantation could restore the lost organic C and N. It was observed that although C and N contents increased after cropping, with increases in all fractions (especially the more labile fPOM fraction), even after up to 39 y, C content within all fractions had not been restored to those observed in the

remnant vegetation soil. These data show that whilst soil can be used as a NET to mitigate climate change, even when soil is removed from productive agriculture, sequestering C and restoring it to the pre-management levels is difficult, especially when humans are actually demanding that global food production from soil must be increased.

**6 Author contributions**

P.M.K., R.C.D., and Z.H.W. designed the experiments and L.H. and M.B. carried them out. L.H. drafted the manuscript with contributions from all authors.

**7 Code and Data availability**

The data and analyses that support these findings will be made available in response to a reasonable request but are not hosted

in an online repository at this time in order to protect the privacy of growers.

**8 Supplement**

The supplement related to this article is available online at: Xxxxxx

**9 Competing interests**

The authors declare that they have no conflict of interest.

## 10 Disclaimer

Publisher's note: Copernicus Publications remains neutral with regard to jurisdictional claims made in the text, published maps, institutional affiliations, or any other geographical representation in this paper. While Copernicus Publications makes every effort to include appropriate place names, the final responsibility lies with the authors.


## 11 Acknowledgements

Part of this research was undertaken on the Soft X-ray spectroscopy (SXR) beamline and the Infrared microspectroscopy (IRM) beamline at the Australian Synchrotron, part of ANSTO (grant numbers AS222_SXR_18487 and AS222_IRM_18449). We thank the beamline scientists, Dr Bruce Cowie and Dr Lars Thomsen, for their technical support on the NEXAFS analysis

and Dr Mark Tobin, Dr Annaleise Klein and Dr Jitraporn (Pimm) Vongsvivut for their technical support on the IRM analysis.

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
