# Peer review of "The influence of land use and management on the behaviour and persistence of soil organic carbon in a subtropical Ferralsol"

_EGUsphere, 2023_

## Author Response (AR1)

**Reviewer 1**

The authors explored the effects of land use changes on a Ferralsol in Australia, focusing on soil organic carbon (SOC) levels and the composition of various fractions in the topsoil. The study is innovative and provides some interesting approaches. However, there are uncertainties concerning the experimental setup that need clarification before publication.

*Major:*

*The description of the study site is insufficient.*

*The dominant vegetation under "remnant vegetation" is unclear and should not be left to the reader's imagination.*

*The details regarding the location of plots and randomization are lacking. Precise information about the location of the sites, for example, with a site plan, is crucial. This is particularly important as the study is assumed to involve pseudoreplicates rather than true replicates. There has been inadequate discussion about the potential overlay of site-specific factors unrelated to land-use changes on the results. While the presence of pseudoreplicates should not serve as a direct exclusion criterion, it needs to be explicitly clarified. For instance, data from before the land-use change could be employed to demonstrate the comparability of sites, or the variance within the plots could be compared to the variance of the treatments (mixed-effects models).*

*The current data status of the study resembles a black-box system. It is not evident to the reader to what extent interim land-use changes in the plantation and grazed pasture sites influence the presently measured values. Consequently, the evaluation, without presenting additional data, is highly speculative. The data provide only a snapshot without insights into historical trends. The terminology used in the analysis is misleading, as it remains uncertain whether SOC increased or decreased due to the land-use change. The reader lacks information on initial or interim contents. There might be a climate change-induced baseline decrease, but its extent varies across systems. Without additional data or detailed descriptions, focusing on the effects of land-use changes in the study lacks coherence. While the data are undoubtedly valuable and interesting, they need to be situated in the appropriate context.*

*Minor:*

*29: This generalization does not simplify but rather complicates understanding.*

*45: Is the citation of the study by Stockmann et al. accurate here?*

*130-145: What values were utilized for pre-edge and post-edge normalization? It would be beneficial to include fitted spectra in the supplements along with fitting statistics. A reference to a study providing detailed information about the beamline would be helpful. What value was graphite calibrated to?*

*152: Is "transmission" the correct term instead of "transition"?*

*165: Were the ANOVAs checked for the normal distribution of residuals and homogeneity of variances? Additional graphs for residual inspection in the supplements would be helpful.*

[Figure]

Table 2: The terms "phenols" and "aromatics" lack adequate definition. Are aromatic compounds restricted to molecules with benzene rings, or do they include other compounds with conjugated double bonds? In principle, phenols are also considered aromatics.

350-356: How can you confirm that the decreased microbial activity is specifically related to soil organic carbon (SOC) and not to other management effects associated with cropping, such as the use of pesticides?

We thank the reviewer for their time and for assisting us improve the manuscript through their constructive criticism. (All line numbers provided refer to the manuscript with Track Changes enabled).

1. We have now provided a schematic diagram to illustrate the site plan (Figure 1). Note that the site is located on private property and so to provide specific information regarding its precise location would be a breach of privacy for this landholder.
2. The reviewer asks to provide more information on the study site. We currently provide information on the climate, land management history (including fertilizer usage), and vegetation (see Methods and Supplementary Table S1). If the reviewer has additional specific information that should be provided, we can do so.
3. We agree that the study uses pseudoreplicates rather than true replicates, and we have modified the text to explicitly state this (Line 489-490).
4. The reviewer expresses concern regarding variability between the various land uses in this paired-site comparison. However, we are unable to provide data from this site from before land-use change (i.e. from 72 y ago) as this information does not exist and obviously cannot be obtained retrospectively. We note that our approach (space-for-time substitution) is a standard approach with well-documented strengths and weaknesses (for example, see Pickett 1989; Space-for-Time Substitution as an Alternative to Long-Term Studies. 110-135). Whilst space-for-time substitution studies have limitations, they are useful for examining temporal trends from different-aged samples. We have now revised

the manuscript to include details of Space-for-Time Substitution studies, including their strengths and weaknesses (Line 490-496). In addition, the reviewer makes an excellent suggestion regarding the use of mixed-effects models to examine the variance within the plots compared to the variance of the treatments, and we have revised the manuscript accordingly (Section 2.7).

5. In a similar manner, we agree with the reviewer that it would be useful to present data from when the plantation and pasture sites were converted into their current land uses (i.e. data from 33-39 y ago). However, again, this information does not exist and, unfortunately, we cannot obtain it as no data were obtained on interim soil properties 33-39 y ago.

6. The reviewer notes that there is a lack of information regarding climate change-induced effects. Although we do not have information for this specific site, there is information published for other sites in Queensland (Australia) that we now cite (Line 497-498).

7. Comment regarding Line 29: We have deleted this information regarding the Soil Carbon 4 Per Mille Initiative avoid confusion (Line 30-34).

8. Comment regarding Stockmann et al on Line 45: We assume that the reviewer is referring to Line 43 rather than Line 45. The reviewer is correct, and this is the wrong citation. Rather, the citation for "Lehmann et al 2020b" should be moved to the end of the sentence, with this now corrected.

9. Comment regarding Lines 130-145: We have now included greater detail for the beamline, the NEXAFS analyses (including pre- and post-edge normalization), and the peak fitting (Section 2.5). We have also included fitted spectra in the Supplementary Information as suggested.

[Figure]

10. Comment regarding Lines 152: The reviewer is correct, and we have changed "transition" to "transmission" (Line 168).
11. Comment regarding Line 165: We have now modified the text to state that the ANOVAs were checked for normal distribution (Line 189).
12. Comment regarding Table 2: Table 2 has been updated as suggested.
13. Comment regarding Lines 350-356: The reviewer raises an important point, and the text has been modified accordingly (Line 385-389).

**Reviewer 2**

*Hondroudakis et al. study the impact of long-term land use change on SOC and the underlying factors regulating its persistence by examining both the molecular diversity and spatial heterogeneity of SOC in a Ferralsol from southeast Australia. Undisturbed soil, cropped soil, pasture, and plantation soils were compared by soil fraction and spectroscopic analysis. They found that soils under >70 years of cropping had lost greater than 70% of their C and N contents relative to soils under remnant vegetation and that the occlusion of POM within soil aggregates is a likely mechanism for SOC persistence.*

*Overall, this ms provides interesting insights into the potential aggregate SOC protection mechanisms in subtropical environments in the context of land use changes. It is well-structured and well-written with informative graphics. However, the fact that the cropped soils had fertilizer additions but other soils didn't is cause for concern about the ability to compare the C and N in those soils. This coupled with the low replication of 5 soils in only the 0-10 cm depth per land use raises questions about how well the samples represent their respective populations and how well the results might generalize. At the very least, these should be stated as limitations to the study in the discussion section.*

*Specific comments*

*30-31 Better not to mix these terms; either pick one and use it throughout or use each appropriately.*

*114-115 Was the soil slurry shaken in SPT to separate the floating POM? This detail is important for interpretation.*

*115-116 Does sonification only separate the aggregate-occluded POM? Does it have the potential to disrupt aggregates in other ways so that the MAOM fraction is then not truly representative of the original MAOM fraction?*

*191 State sample size again.*

*318 Were decreases for N in other studies of similar magnitude? If so, explicitly state that.*

*377 - 378 Or the very fast turnover of POM in this subtropical climate.*

*Technical corrections*

*102-104 Remove paragraph space to combine the two paragraphs.*

*183 reference table 1 with the first mentions of data.*

We appreciate the time that the reviewer has taken to read our manuscript and provide us with this valuable feedback.

1. The reviewer raises an important point about the number of replicates used (five). We consider this to be an adequate number and so we disagree with the concerns that the reviewer raises in this regard. Regardless, to address the concerns of the reviewer, we have modified the Discussion (line 489-496).
2. The reviewer appears concerned that (limited) fertilizers were applied to the cropping soil. However, we disagree that is of concern – our study aimed to examine the effect of land-use change on soil C, and the application of fertilizer is typical practice for these cropping systems. Thus, understanding the effect of the cropping system on soil properties requires implementation of those normal management practices. Regardless, we have modified the text to clearly highlight this point (Line 503-508).
3. Comment regarding Lines 30-31: As suggested by the reviewer, we have now separated the terms 'SOC' and 'SOM'.
4. Comment regarding Lines 114-115: The soil was not shaken in the SPT slurry, and the sentence has been updated accordingly (Line 120-121).
5. Comment regarding Lines 115-116: This is an important point and we have now included information accordingly (Line 122-124). Specifically, a preliminary study was undertaken to ensure that the energy input during sonication was the minimum required to ensure recovery of the POM, with this being crucial in minimizing sonication-induced artefacts. The energy input (400 J/mL) is within the range commonly used (for example, Steffens et al 2009 EJSS 60: 198-212) and values of 500 J/mL are common for reactive soils (for example, Spielvogel et al 2007; Asano and Wagai 2014; Heckman et al 2013). However, given that occlusion in aggregates and surface sorption occur in a continuum (Kögel-Knabner et al. 2008 JPNSS 171:61-82), there is no single ideal approach.
6. Comment regarding Line 191: We have now stated the sample size in the caption for Table 1 as suggested.
7. Comment regarding Lines 318: We have modified this sentence to explicitly state that the magnitude is similar.
8. Comment regarding Lines 377-378: We agree with the reviewer and have modified this sentence to note that it could also be related to the fast turnover of POM in this subtropical climate (Line 411).
9. Comment regarding Lines 102-104: We have combined these two paragraphs as suggested.
10. Comment regarding Lines 183: We have cross-referenced Table 1 as suggested (Line 205).